# Deep Brain Stimulation of the Posterior Insula in Chronic Pain: A Theoretical Framework

**DOI:** 10.3390/brainsci11050639

**Published:** 2021-05-15

**Authors:** David Bergeron, Sami Obaid, Marie-Pierre Fournier-Gosselin, Alain Bouthillier, Dang Khoa Nguyen

**Affiliations:** 1Service de Neurochirurgie, Université de Montréal, Montréal, QC H3T 1L5, Canada; sami.obaid@umontreal.ca (S.O.); marie-pierre.fournier-gosselin@umontreal.ca (M.-P.F.-G.); alain.bouthillier@umontreal.ca (A.B.); 2Service de Neurologie, Université de Montréal, Montréal, QC H3T 1L5, Canada; d.nguyen@umontreal.ca

**Keywords:** deep brain stimulation, insula, chronic pain, neuromodulation

## Abstract

Introduction: To date, clinical trials of deep brain stimulation (DBS) for refractory chronic pain have yielded unsatisfying results. Recent evidence suggests that the posterior insula may represent a promising DBS target for this indication. Methods: We present a narrative review highlighting the theoretical basis of posterior insula DBS in patients with chronic pain. Results: Neuroanatomical studies identified the posterior insula as an important cortical relay center for pain and interoception. Intracranial neuronal recordings showed that the earliest response to painful laser stimulation occurs in the posterior insula. The posterior insula is one of the only regions in the brain whose low-frequency electrical stimulation can elicit painful sensations. Most chronic pain syndromes, such as fibromyalgia, had abnormal functional connectivity of the posterior insula on functional imaging. Finally, preliminary results indicated that high-frequency electrical stimulation of the posterior insula can acutely increase pain thresholds. Conclusion: In light of the converging evidence from neuroanatomical, brain lesion, neuroimaging, and intracranial recording and stimulation as well as non-invasive stimulation studies, it appears that the insula is a critical hub for central integration and processing of painful stimuli, whose high-frequency electrical stimulation has the potential to relieve patients from the sensory and affective burden of chronic pain.

## 1. Introduction

Remarkable progress has been made in understanding the biological, cognitive, and psychological bases of pain. Chronic pain not only is a detriment to the quality of life but also leads to a loss of contribution to and integration into society. Most pain medications are efficient for short-term nociceptive pain but have significant cognitive and emotional side effects when taken in the long term. Specifically, chronic opioid use often leads to narcodependence, paradoxical pain hypersensitization, and chronic depression [1]. Despite our increasing understanding of the neurobiology of pain, current pharmacotherapy fails to relieve the suffering of many patients with chronic pain, and there have been few advances in this field in recent decades.

Neurosurgeons have long attempted to prevent the suffering of patients with chronic pain by disrupting various structures along pain pathways, including peripheral nerves, dorsal roots, the spinothalamic pathway in the spinal cord, the midbrain, the thalamus, and the cingulate cortex, by lesioning, electrically stimulating, or perfusing with analgesic/anesthetic agents [2]. Initially, ablative surgeries such as cordotomy, midline myelotomy, thalamotomy, and cingulotomy have been carried out to disrupt ascending pain signals. Ablative surgeries have been mostly replaced by high-frequency electrical stimulation due to the high risk of unwanted, permanent neurologic deficits, the unadjustable degree of pain control, and the risk of neuropathic pain or anesthesia dolorosa [2,3,4]. Notably, epidural spinal cord stimulation is now a standard procedure for the treatment of neuropathic pain or failed back surgery syndrome [2,3,5,6,7,8,9,10]. Although currently considered an off-label use by the Food and Drug Administration (FDA), as two multicenter trials demonstrated limited efficacy [11], some centers still offer deep brain stimulation (DBS; usually of the thalamus or periaqueductal gray) to patients with severe refractory pain [4,12,13]. 

DBS targets for chronic pain were initially chosen based on the classical understanding of pathways involved in acute nociceptive pain, with second-order ascending neurons from lamina I-V of the spinal dorsal horn synapsing in the ventral posterior lateral and medial nuclei (VPL/VPM) of the thalamus before reaching the somatosensory cortex, and descending pain modulation neurons originating from the periaqueductal gray (PAG) as part of the endogenous opioid system [4]. High-frequency electrical stimulation of these regions and their vicinity elicited immediate anesthesia in animal models and in early human clinical trials but could never convincingly deliver sustained pain relief in clinical trials involving patients with refractory chronic pain. Decades of research allowed us to understand that chronic pain engages complex networks far beyond the classical pain pathways, networks underlying the central hypersensitization to stimuli. and dysregulation of cognitive-emotional responses [14,15,16]. We hypothesize that these regions would represent more promising DBS targets to elicit sustained relief from chronic pain.

Multiple lines of evidence have allowed us to identify the insula as an important central integration center for painful stimuli, potentially representing the interface between sensory and limbic systems during painful experiences. First, neuronal tracing of pain pathways in primates has identified the posterior insula as an important relay in the spinothalamic pathway through connections from the VPM nucleus of the thalamus [17,18,19]. Second, functional imaging of individuals undergoing painful experiences has consistently shown activation of the posterior insula, with high correlation with pain intensity [20,21,22,23]. Third, functional imaging of patients with chronic pain syndromes has consistently identified abnormal connectivity and neurotransmitter levels in the posterior insula [24,25,26,27,28,29,30,31,32,33,34]. Fourth, brain lesions involving the insula have been shown to elicit a syndrome of asymbolia for pain, an indifference to painful stimuli despite the preservation of sensory discrimination [35,36,37]. Fifth, electrode implantation in the insula for invasive monitoring of refractory epilepsy has revealed that low-frequency stimulation of the posterior insula can elicit pain, and its high-frequency stimulation increases pain thresholds [38,39,40,41,42,43,44]. 

In this manuscript, we present a narrative review of the different lines of evidence linking the insula to the experience of pain and introduce the insula as a novel anatomical target for DBS in patients with chronic pain. 

## 2. Neuroanatomical Studies

The textbook description of ascending pain pathways involves three synapses. The initial painful stimuli (of various natures: thermal, mechanical, chemical) are detected by nociceptor ion channels and transmitted by fast-speed myelinated A-delta fibers and low-speed unmyelinated C fibers (first-order neurons) to lamina I to V of the superficial spinal dorsal horn. After synapsing in lamina I-V, second-order neurons decussate in the spinal cord, ascend through the anterolateral spinothalamic tract, and ultimately synapse with third-order neurons in the ventral posterior lateral and medial nuclei of the thalamus (VPL/VPM). Third-order neurons then synapse in the postcentral gyrus, also known as the primary somatosensory cortex. Recent neuronal tracing studies in primates have shown that lamina I neurons, constituting approximately half of the ascending spinothalamic fibers in humans and playing a crucial role in the transmission of painful and/or unpleasant interoceptive stimuli [45,46,47,48], project to a specific thalamic nucleus (the posterior part of the ventral medial nucleus of the thalamus [VMPo]) [45,48], which, in turn, projects to the fundus of the superior limiting sulcus (SLS) of the insula rather than the classical primary somatosensory cortex [18]. This new pathway defines the dorsal fundus of the insular cortex as the primary cortical center for pain and interoception [18,49]. The anterior area and the posterior area of the fundus of the insular cortex have been named the primary interoceptive cortex, receiving vagal and spinal interoceptive afferents from the solitary tract nucleus and lamina 1 of the spinal cord, via the basal and posterior parts of the ventral medial nucleus of the thalamus (VMb and VMpo), respectively [49,50]. Microelectrode recordings from barbiturate-anesthetized macaque monkeys have shown that neurons in the dorsal fundus of the posterior insula respond to innocuous and noxious thermal stimuli as well as to noxious mechanic pinch stimuli, with a somatotopic representation of the foot, hand, and face from posterior to anterior [51]. In addition to constituting an important relay station in ascending pain pathways, the posterior insula is densely connected with the anterior insula, which projects to various brainstem nuclei involved in the descending pain modulation system (central raphe, locus coeruleus, ventral tegmental area, and periaqueductal gray) and multiple limbic structures involved in attributing emotional valence to the painful stimulus, such as the anterior cingulate cortex (ACC) and the amygdala [49,50,52]. In mice, optogenetic activation of the projection between the posterior insula and dorsal raphe promoted pain hypersensitization and maintained nociceptive hypersensitivity, even in the absence of a nociceptive drive [53]. Conversely, optogenetic inhibition of the pathway prevented hypersensitization. Altogether, these recent advances regarding the integration of the posterior insula in the networks of pain perception provide a strong neuroanatomical framework for potential DBS of the posterior insula as a treatment for chronic refractory pain. 

## 3. Brain Lesion Studies 

In 1928, Schilder and Stengel presented a patient with sensory aphasia, self-mutilative behavior, and an intriguing indifference to painful stimuli [35]. The patient could distinguish between sharp and dull stimuli equally throughout the body but was unable to recognize the unpleasant component of a painful stimulus. This condition—named asymbolia for pain—remains a scarcely reported consequence of localized brain lesions, with about 30 reported cases to date [35,54,55,56,57,58,59]. In case reports of asymbolia for pain, patients frequently had lesions encompassing the dominant inferior parietal lobe, frontoparietal cortex, second somatosensory area, parietal operculum, and adjacent insular cortex [35]. In his famous “Disconnexion Syndromes in Animals and Man,” Norman Geschwind suggested that asymbolia for pain represents a disconnection syndrome between the secondary sensory cortex and the limbic system [36,37]. He hypothesized that a lesion of the posterior insula could interrupt connexions between the secondary somatosensory cortex and the amygdala, hence blunting the emotional response to pain without affecting the perception of the stimuli per se [36,37]. In 1988, Berthier et al. presented six patients with asymbolia for pain. The cardinal feature was a striking attenuation of the emotional and behavioral response to pain, although all patients could perceive and cognitively identify the painful nature of a stimulus [35]. All patients had lesions encompassing the insula, and one patient had a discrete ischemic lesion in the posterior insula and parietal operculum, strongly suggesting that the involvement of such structures may be sufficient for the production of the syndrome [35]. 

Interestingly, Isnard et al. reported the occurrence of painful ictal attacks in a patient with small cortical dysplasia located in the posterior third of the right insula [60]. Investigation with intracerebral electrodes revealed high-frequency (>40 Hz) activity of the posterior insula, midcingulate cortex, and parietal operculum (but not the other 14 implanted regions) directly preceding the clinical expression of the seizure. The seizure was characterized by a painful facial expression, accompanied by a scream and subjectively by a feeling of intense continuous burning over the left half of the body, sparing the head, with maximal intensity for a few seconds and progressive weaning over one or two minutes. Intracortical bipolar electrical stimulation of the posterior insula was able to reproduce painful somatosensory symptoms with similar quality as the initial pain of spontaneous seizures, with a more restricted and slightly different topography. After thermocoagulation of the posterior insular cortical dysplasia, the patient remained free of pain attacks. The same group further described five patients with painful somatosensory seizures originating from the operculo-insular cortex [61]. In four patients, a syndrome of ictal symbolism for pain—defined as pain behavior without declarative pain sensation despite fully preserved contact and vigilance—was observed in four patients with focal seizure discharges within the posterior operculo-insular cortex and little propagation to other cortical structures [62]. 

Furthermore, in 2010, Garcia-Larrea et al. described a novel central pain syndrome called “operculo-insular pain” [63]. In their series, they identified five patients with stroke involving the posterior insula and inner parietal operculum, who presented with central pain and pure thermoalgesic sensory loss. They showed profound alteration of thermal and pain thresholds, with preservation of basic lemniscal modalities (discriminative touch, joint position sense, stereognosis. and graphaesthesia) [63]. 

These clinical observations place the posterior insula as a crucial hub for the integration of sensory and affective dimensions of pain. 

## 4. Functional Imaging Studies 

The cortical and subcortical areas involved in pain processing have long been studied with functional MRI (fMRI) and positron emmision tomography (PET). In a study involving a total of 114 participants, Wager et al. (2013) developed an fMRI-based measure that predicted pain intensity at the level of the individual [20]. The pattern included the bilateral dorsal posterior insula, the secondary somatosensory cortex, the anterior insula, the ventro-lateral and medial thalamus, the hypothalamus, and the dorsal anterior cingulate cortex, referred to as a pain matrix [20]. This study informed us about the involvement of this group of regions in pain processing, but their respective role in pain integration remained debated. Segerdahl et al. (2016) exploited arterial spin-labeling quantitative perfusion imaging and a newly developed procedure to identify a specific role for the dorsal posterior insula in pain [21]. In 17 subjects, the only significant positive correlation between absolute cerebral blood flow changes and pain ratings within subjects was observed in the contralateral dorsal posterior insula [21]. In 2000, Craig et al. had also demonstrated, using positron emission tomography (PET), that graded cooling stimuli correlated with contralateral metabolic activity in the dorsal margin of the middle/posterior insula [64]. 

These findings in healthy subjects were further validated in clinical populations with various chronic pain syndromes. An extensive literature has highlighted that chronic back pain disrupts default-mode network activity [65,66] and is associated with abnormal functional connectivity of the insula [67,68]. In fibromyalgia, multiple lines of evidence link to anatomical and functional abnormalities in the anterior and posterior insula. With the advent of fMRI studies, patients with fibromyalgia have been consistently found to exhibit higher activity in the contralateral insular cortex in response to painful stimuli [26,69,70]. Patients with fibromyalgia also exhibit an increased glutamate/creatine ratio in the insula on proton magnetic resonance spectroscopy (H-MRS), reinforcing the idea that neural activity is augmented within this region in fibromyalgia [26,31,32,33,34]. Finally, numerous studies have highlighted abnormal resting-state connectivity of the insula in patients with fibromyalgia, correlated with pain intensity and modulated by treatment [24,25,27,28,29,71,72]. More specifically, the posterior insula of patients with fibromyalgia exhibits increased connectivity with components of the default-mode network and decreased connectivity with sensorimotor areas [27,72], and the anterior insula exhibits increased connectivity with the anterior cingulate cortex and parahippocampal gyrus [28]. Similar associations were found in other chronic pain syndromes such as chronic pelvic pain [73,74,75], complex regional pain syndrome [76], temporomandibular disorder [77], and chronic knee osteoarthritis [78]. 

Integrating these imaging findings with anatomical studies, we can hypothesize that abnormal insular activity and connectivity mediates the transition from acute to chronic pain through central pain hypersensitization (posterior insula–dorsal raphe–descending pain facilitation [53]) and maladaptive plasticity in mesocorticolimbic reward/motivational circuitry (anterior insula–anterior cingulate cortex–prefrontal cortex–nucleus accumbens [14,79]) strengthening emotional and affective pain mechanisms [16]. Reversing abnormal insular activity and connectivity through high-frequency electrical stimulation might restore maladaptive pathways mediating sensory and affective hypersensitivity in patients with chronic pain. 

## 5. Intracranial Recordings during Painful Stimuli

Intracranial recordings of brain activity during painful stimuli have also helped uncover the role of the insula in central nociceptive integration. Single-unit recordings in monkeys have demonstrated the existence of nociceptive neurons in SII and the insula in the early 1980s [80,81]. In humans, intracranial recordings have shown that the earliest cortical response to painful laser stimulation occurs with a 180–230 ms latency in the posterior insula and SII [82,83,84,85]. Further studies have shown that SII responses are able to encode the intensity of laser thermal stimuli from the sensory threshold up to the pain threshold level but tend to show a ceiling effect for increasing pain, while the posterior insular cortex fails to detect stimulus intensity changes for low levels of stimulation (around the sensory perception threshold) but encode stimulus intensity variations in the painful range without showing saturation effects for the highest painful intensities [82]. Nociceptive responses in the posterior part of the insula were recorded with the shortest latencies and the highest amplitudes. In contrast, they peaked later and with smaller voltages in the anterior insular cortex, suggesting that nociceptive input is first processed in the posterior insula, where it is known to be coded in terms of intensity and anatomical location, and then conveyed to the anterior insula, where the emotional reaction to pain is elaborated [42,86,87]. Based on intracranial recordings showing identical onset latencies of amygdalar and insular responses to painful stimuli, Bastuji et al. suggested that these regions process sensory and affective components of pain in parallel through the spino-thalamo-cortical and spino-parabrachial pathways, respectively [86]. Analyzing evoked field potentials and coherence analyses, Bastuji et al. recently suggested that sensory and affective inputs from the posterior insula and amygdala, respectively, converge in the anterior insula, integrating sensory with emotional input and hence formulate an experience of pain [87]. 

## 6. Electrical Stimulation of the Insula in Humans

Wilder Penfield and his colleagues at the Montreal Neurological Institute were the first to use direct cortical stimulation for intraoperative cortical functional mapping in the context of epilepsy surgery [88]. In a famous study, Penfield was able to elicit positive responses from the stimulation of a total of 82 separate points on the insular cortex in 36 patients during awake epilepsy surgery [89]. The majority of these responses were divided into two distinct groups: gastric sensory or motor phenomena were the result of 32 stimulations (epigastric sensation, nausea, borborygmus, chewing, salivation, strange taste), and somatic sensations were the result of 30 stimulations (motor movements; feelings of tingling; numbness; warmth in the lips, tongue, mouth, and throat; contralateral hand or finger, legs, etc.). Since these landmark findings, many groups have reported various responses to electrical stimulation of the insula, mostly during the presurgical evaluation of refractory epilepsy [38,39,41,43,44,80,90,91,92,93,94,95,96,97,98]. Mazzola et al. (2012) analyzed 4160 cortical stimulations from 164 consecutive patients undergoing intracortical depth recordings using stereo-electroencephalographic (SEEG) procedures for presurgical evaluation of medically refractory epilepsy [39]. A painful somatic sensation was evoked by cortical stimulation in only 60 of the 4160 stimulated sites (1.4%), which were concentrated in the medial part of the SII area or in the posterior and upper part of the insular cortex. Further, the voltage required to elicit a painful sensation was lowest at the posterior aspect of the posterior insula and progressively increased with more anterior stimulations. Non-painful paraesthesia represented 35% of responses to insular stimulation (*n* = 151) and was described as tingling, a feeling of pulsation, a feeling of vibration, or a feeling of numbness or unpleasant non-painful paresthesia such as pins and needles or a slight electric current. Afif et al. (2010) studied 25 patients implanted with SEEG in the insula for presurgical evaluation of medically refractory epilepsy: eight responses (8/83 of all responses evoked in the insula, all located in the upper portion of the middle short gyrus of the insula) were described as painful, including headache, throat pain, or repetitive pinprick sensations [44]. Other responses included sensory (paresthesias and localized warm or cold sensations), motor, auditory, oropharyngeal, speech disturbances (including speech arrest and reduced voice intensity) and neurovegetative phenomena (facial reddening, hypogastric sensations, anxiety attacks, respiratory accelerations, sensations of rotation, nausea) in a somatotopy concordant with Mazzola et al.’s [98] and Penfield’s initial descriptions [89]. 

Finally, Denis et al. (2016) studied six subjects implanted with depth electrodes for presurgical evaluation of their medically refractory epilepsy [43]. They evaluated the impact of prolonged (10 min) high-frequency (150 Hz) insular stimulation on sensitivity to pain by hot, cold, and pressure stimuli. They found that high-frequency insular stimulation increased the heat pain threshold on the ipsilateral (*p* = 0.003; *n* = 6) and contralateral sides (*p* = 0.047; *n* = 6). The pressure pain threshold was not modified by insular stimulation (ipsilateral: *p* = 0.1123; contralateral: *p* = 0.1192; *n* = 6). Five of the six subjects enrolled in the study had a cortical resection that included the insula because of epileptic spiking in this region. Two of the three subjects who had a posterior operculo-insulectomy developed a postoperative central pain syndrome associated with a contralateral loss of thermal sensibility. These preliminary results suggest the potential benefit of high-frequency DBS of the posterior insula and its vicinity in patients with refractory chronic pain. 

## 7. Non-Invasive Insular Stimulation in Patients with Chronic Pain

Clinicians and researchers have long attempted to relieve chronic pain using non-invasive stimulation, either transcranial direct current stimulation (tDCS) or repetitive transcranial magnetic stimulation (rTMS) [99,100]. The most studied targets have been the primary motor cortex (M1), primary sensory cortex (S1), and the dorsolateral prefrontal cortex (DLPFC), as they are superficial and hence easily accessible through transcranial stimulation [100]. We historically considered that the insula and other deep structures could not be selectively modulated by transcranial, non-invasive techniques. Recently, some groups have claimed to achieve selective stimulation of deeper cortical regions using cooled double-cone coils [101,102], although the effectiveness of the technique remains debated [103]. Based on this advance, Lenoir et al. showed that deep continuous theta burst stimulation (cTBS) of the right operculo-insular cortex in 17 patients selectively impaired the ability to perceive thermo-nociceptive input conveyed by Aδ-fiber thermo-nociceptors without concomitantly affecting the ability to perceive innocuous warm, cold, or vibrotactile sensations [104]. During this study, two patients experienced short-lasting manifestations compatible with a partial TMS-induced seizure—specifically, these patients experienced breathing difficulties associated with laryngeal sensation and thoraco-abdominal heaviness, followed by a dystonic posture of the left or right hemibody [104]. Galhardoni et al. recently performed a randomized, double-blind, sham-controlled, three-arm parallel study comparing the analgesic effects of stimulation of the anterior cingulate cortex (ACC) or the posterior superior insula (PSI) against sham deep repetitive TMS (dTMS) in 98 patients with central neuropathic pain. The heat pain threshold significantly increased after treatment in the PSI-dTMS group from baseline compared to sham-TMS but had no significant effect on pain interference with daily activities, neuropathic pain symptoms, and the quality of life [105]. Overall, given the small risk of seizures, the difficulty to selectively stimulate specific posterior insular subregions with non-invasive techniques, Zugaib et al. suggested that the cost–benefit ratio for testing the new protocols for neuromodulation of the operculo-insular cortex using rTMS seemed unfavorable [106].

## 8. Previously Studied DBS Targets for Chronic Pain

DBS was first attempted to treat chronic pain more than 50 years ago. Olds and Milner described in 1954 experiments with electrical stimulation of the septal area in rats, which led to a rewarding effect and compulsive self-stimulation [107]. These results led Heath and Mickle to attempt electrical stimulation of the septal area in patients with schizophrenia. Stimulation made patients feel alert and well, with instances of immediate relief from chronic pain in a few cases [108]. These results could not be reproduced by Gol (1966), who achieved pain control with electrical stimulation of the septal area in only one of six patients [108], and this target was mostly abandoned. In 1969, Reynolds showed that electrical stimulation of the midbrain central gray could induce complete analgesia in rats [109], which was reproduced by Mayer et al. (1971) [110]. These findings led Richardson et al. (1977) to attempt periaqueductal gray electrical stimulation in five patients scheduled to undergo thalamotomy for chronic pain. Electrical stimulation elicited a reduction of chronic pain and hypoalgesia to a pinprick but also produced numerous undesirable side effects, including nystagmus, nausea, vertigo, and a feeling of a “rising vapor” [111]. The ventral medial nucleus of the thalamus was also stimulated in some patients, producing “good-to-very-good hypalgesia” [111]. Subsequently, Richardson et al. (1977) implanted electrodes in the periventricular gray region for chronic self-stimulation in patients with chronic pain, with successful pain control in 6/8 patients [112]. Analgesia was obtained with low-frequency, low-amplitude current, with a higher frequency and current producing noxious stimuli [112]. Periaqueductal gray DBS is believed to inhibit pain by activating the endogenous opiate system, as its effect was shown to be reversed by an injection of naloxone, an opioid antagonist [113,114,115]. Inspired by the success of thalamotomy for chronic pain [116,117], Mazars et al. (1973, 1974, 1975) performed electrical stimulation of the nucleus ventralis posterolateralis of the thalamus, with pleasant paresthesias supplanting painful sensations in patients with chronic pain [118,119,120]. The ventral posterolateral nucleus and ventral posteromedial nucleus (VPL/VPM) subsequently became accepted DBS targets for refractory pain. Likewise, Spooner et al. (2007), inspired by the efficacy of cingulotomy for the treatment of chronic pain, attempted anterior cingulate cortex (ACC) DBS for the treatment of the affective component of refractory pain [121]. Aziz et al. subsequently treated 24 chronic pain patients with bilateral ACC DBS [12,122,123,124,125]. Unlike stimulation of conventional targets of DBS for pain (VPL/VPM of the thalamus, periaqueductal gray matter), which reduce the intensity of pain, ACC stimulation was shown to decrease its affective component with significant improvement in the quality of life; patients stated that they continued to have pain, but it was “not distressing,” “not particularly bothersome,” or “not worrying anymore” [123]. Four patients experienced recurrent seizures during the follow-up period, with persistent epilepsy despite the cessation of the stimulation in two patients [123,126]. Other less studied DBS targets for chronic pain include the centro-median and parafascicularis nuclei of the thalamus [127], the posterior limb of the internal capsule [128], and the ventral striatum/anterior limb of the internal capsule (VS/ALIC) [129]. Recently, a randomized, double-blind, crossover trial of VS/ALIC DBS in 10 patients with refractory neuropathic central pain was shown to improve clinical outcomes related to the affective sphere of chronic pain. 

To date, 22 studies of DBS for pain have been published [4], 17 of which used VPM/VPL or periaqueductal gray matter as targets (200 patients), and the remainder ACC (24 patients), VS/ALIC (10 patients), centro-median and parafascicularis nuclei of the thalamus (3 patients), and PLIC (3 patients). For PAG/PVG or VPM/VPL targets, most patients were recruited through a prospective study in Oxford (85 patients implanted, 59 with follow-up data) [130], two multicenter studies in the U.S. (246 patients implanted, 40 patients with follow-up) [11], a study in Germany (54 patients implanted, 32 patients with follow-up data) [131], and a study in Saskatchewan, Canada (68 patients implanted, 53 with follow-up data) [132]. Two multicenter trials of DBS for pain were conducted to seek US Food and Drug Administration approval, the first in 1976 by using the Medtronic model 3380 electrode (196 patients) and the second in 1990 with the model 3387 (50 patients). Neither trial satisfied the efficacy criteria of at least half of patients reporting at least 50% pain relief 1 year after surgery [11]. However, the considerable loss of patients at follow-up resulted in a steady increase with time in the proportion of patients with 50% pain relief; 2 years after implantation, they comprised 18 of the 30 remaining patients (60%) followed up in the model 3380 trial and 5 of the 10 in the model 3387 trial (50%). In the Oxford cohort, 59/85 of patients retained implants 6 months after surgery, and 39/59 (66%) of those implanted presented a sustained global improvement of their EuroQol-5 dimensions (EQ-5D). In the German cohort, the best long-term results were elicited among patients who suffered from failed back surgery syndrome; over a mean follow-up of 3.5 years, all but one patient improved by at least 50%. In contrast, only 2 of 11 patients with post-stroke pain had the lead permanently implanted, while other patients had their lead explanted [131]. In the Saskatchewan cohort, 53 of 68 patients (77%) elected internalization of their devices; 42 of the 53 (79%) continued to receive adequate relief of pain; patients with failed back syndrome, trigeminal neuropathy, and peripheral neuropathy fared well with DBS, whereas those with thalamic pain, spinal cord injury, and post-herpetic neuralgia did poorly [132]. Multiple studies with smaller cohorts elicited similar results [127,128,133,134,135]. When VPL/VPM is targeted, pleasant paraesthesia supplants a painful sensation. PVG/PAG stimulation induces a sense of warmth, floating, and dizziness at the threshold stimulation with frequencies of 50 Hz and a pulse width of 210 ms. At higher intensities, anxiety or even panic was reported by the patients [131].

## 9. Conclusions

In light of the converging evidence from neuroanatomical studies, brain lesion studies, neuroimaging studies, intracranial recording and stimulation, as well as non-invasive stimulation reports, it appears that the insula is a critical hub for central integration and processing of painful stimuli, whose high-frequency electrical stimulation has the potential to relieve patients from the sensory and affective burden of chronic pain. 

Needless to say, DBS of the insula would raise multiple challenges:
First, the optimal electrode target(s) within the insula and landmarks to achieve precise and consistent implantation will require further investigation. The experience gathered by epilepsy surgeons with SEEG implantation in the insula would certainly facilitate their safe and precise positioning. Furthermore, data gathered from intracranial electrical stimulation and recording in epileptic patients have helped to delineate the primary somatosensory region receiving spinothalamic input from the VMPo, most likely to directly affect pain perception [39,44]. Awake surgery with microelectrode recording during painful stimulation or intracranial stimulation would also help refine electrode positioning in the operating room. It would certainly be worthwhile to consider the implantation of additional electrodes in the middle or anterior insula to modulate the affective components of pain, in addition to its somatosensory components. Second, the selection of the proper patient population represents a significant challenge. As discussed in Section 4, studies have highlighted abnormal insular activity and connectivity in many chronic pain syndromes. However, chronic pain is an extremely heterogeneous condition, and it remains unclear which patients would most benefit from insular DBS. Fibromyalgia is the pain disorder with the clearer neuroimaging evidence pointing to circuit abnormalities centered on the insula, as the source of sensory and affective hypersensitivity to external and interoceptive stimuli [26]. However, the heterogeneity of the syndrome and its psychiatric comorbidities represent major challenges. Patients with fibromyalgia can be expected to be more sensitive to post-operative pain from the insertion of the electrode and battery. Demonstration of cost-effectiveness in a syndrome that carries important negative bias and misunderstanding in the general population and medical community might also prove challenging. Other promising patient populations include central neuropathic pain from stroke or spinal cord damage, complex regional pain syndrome, and post-amputation phantom pain. Whatever the source of chronic pain, patients will need to be managed at a tertiary-care chronic pain clinic, fail all conventional treatments, and have symptoms severe and incapacitating enough to justify the invasiveness and cost of DBS as a last-resort therapy. Third, there is little data from which to infer the potential effects of electrical stimulation of the posterior insula on the activity of pain networks or to infer which stimulation parameters would be most effective. Low-frequency stimulation of the posterior insula was shown to elicit pain [39], and high-frequency stimulation was shown to transiently increase temperature pain thresholds [43]. The exact neural mechanisms underlying these effects are currently unknown. Initially, since the clinical response elicited by high-frequency stimulation was similar to a lesion of the same region (for instance, the subthalamic nucleus), high-frequency stimulation was thought to functionally inactivate the neuronal bodies, from depolarization blockade and/or neurotransmitter depletion [136]. We have since understood that DBS pulses are propagated to anatomically connected regions by orthodromic and antidromic action potentials generated in axons in the vicinity of the target region, including afferents and efferents of the target, as well as nearby passing fibers [136]. DBS also modifies the neurotransmitter environment by its effect on astrocytes [136]. We currently do not have the neuroanatomical or computational knowledge to predict the effect of localized electrical stimulation on the activity of multiregional brain networks, although this is the subject of an ongoing investigation [137]. It is equally difficult to predict which stimulation parameters would better re-establish proper neuronal activity within the insular network. Based on preliminary evidence with high-frequency insula stimulation (150 Hz) [43] and experience with DBS of the ACC (130 Hz, 450 μs) [123], a region with a baseline neurophysiology that is similar to the insula [138], we intend to use an initial stimulation frequency of 130 Hz. The use of a sensing-enabled DBS system will help to better understand the effect of different stimulation parameters on insular neurophysiology. Fourth, regarding the intensity of stimulation, the risk of stimulation-induced seizure is an important concern. The insula is well known to be an epileptogenic structure [139], and non-invasive stimulation of the insula was shown to trigger partial insular seizure in a recent study [104]. DBS of the ACC—another well-known epileptogenic region—has also triggered partial seizures in some patients when stimulation intensity was increased too quickly, including continuing epilepsy after electrode removal in a patient [126]. To lower the risk of stimulation-induced seizure and better refine stimulation intensity throughout the day, the use of a sensing-enabled DBS system would be desirable [140]. Stimulation with the cycle mode, rather than prolonged continuous stimulation, has also been proposed to reduce the risk of seizures [140,141]. Moreover, subjective experience of pain varies throughout the day, and hence, there would be a possibility to modulate stimulation according to the sensing of brain activity—in other words, closed-loop stimulation [142]. In this regard, an important challenge will be to develop reliable, real-time decoders of the subjective pain state based on intracranial recordings. This is the subject of ongoing efforts by different research teams [142]. Fifth, the outcome measures of insular DBS will require thoughtful considerations. Of course, pain ratings on the visual analog scale, quality-of-life ratings, pain interference with daily activities, mood, medication use, and cortical electrophysiology measurements will represent important outcome measures. Appropriate trial designs with sham stimulation groups will be of utmost importance, given the high risk of the placebo effect. Finally, it will be important to monitor the persistence of the effect over time, as previous trials of DBS for chronic pain with other target regions have all shown initially encouraging results, which proved only temporary over time in most patients [4,11]. 

Altogether, we believe that recent scientific advances in the neuroscience of pain suggest that the insula represents a promising DBS target for the treatment of chronic refractory pain. We hope that this manuscript forms the theoretical basis for a phase 1 trial of insular DBS in the future.

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
