# Peer review of "Deep Brain Stimulation of the Posterior Insula in Chronic Pain: A Theoretical Framework"

_brainsci, 2021, doi:10.3390/brainsci11050639_

Round 1

Reviewer 1 Report

This is a well written and extremely interesting review.  I enjoyed reading it.  I have no suggested changes.

Author Response

We thank the reviewer for these comments.

Reviewer 2 Report

The paper is an excellent review of the history of neurosurgical attempts to treat chronic pain and a variety of neurophysiological evidence from a variety of perspectives to suggest that stimulation of the posterior insula, theoretically, should help in the treatment of chronic pain.  The authors have convincingly made the case.

However, there is a serious flaw in the paper.  It is extremely unfortunate that they authors did not apply the same excellent diligent effort to understanding the mechanisms of action of DBS.   The result is the total paper is flawed and naïve even as the clinical features and anatomy were so well discussed.  Indeed, the authors give scant attention to the mechanism of action.  Worse, the presumptions in what they mention about DBS are outdate and unfounded and thus, the paper risks perpetuations of these misconceptions.  In fact, the failure of past attempts of DBS to effectively control chronic pain may have more to do with the naivety of the electroneurophysiology than it does to the actual physiology.  It would be wrong to blame with physiology if scientists and clinicians were naïve in the prescription of electricity.  However, based on the quality of work in what was presented, this reviewer has every confidence that the situation relative to DBS could be easily reconciled.  The authors may find Montgomery, EB Jr., Deep Brain Stimulation Programming: Mechanisms, Principles and Practice, 2nd edition, Oxford University Press, 2017, helpful.

Presumption 1: the effects of DBS can be understood as derivative of the anatomical target stimulated.  This is totally unfounded.  We know from careful neurophysiological studies that it is highly likely that the efficacy of DBS of the subthalamic nucleus (STN) in Parkinson’s disease may have little to do with changes in the neurons of the STN.  Rather the therapeutic mechanisms may have more to do with antidromic activation of axons terminating or merely passing through the vicinity of the STN.  That is why it is a misnomer that there is such a thing as DBS of X, where X is the targeted anatomical structure.  The more appropriate description is DBS in the vicinity of X and all the axons that pass within the vicinity of the DBS cathode.  So what the authors would presume as effects of electrical stimulation of the posterior insula may have very little to do with the posterior insula but rather, stimulation of the posterior insula is a “gateway” to activation of the relevant network in which the posterior insula is situated.  For example, a number of neurophysiological studies have demonstrated antidromic activation from centers that project to the vicinity of the STN, including the contralateral STN and ipsilateral cortex and thalamus.

This is not to say that DBS of the posterior insula is not a reasonable target as it still may allow access to the relevant network.  Further, there may be other features of the posterior insula that convey an advantage to stimulating the posterior insula over other potential gateways in the same networks.  For example, surgical access to the posterior insula may be easier with lower adverse effects compared to stimulation of the thalamus and peri-aqueductal grey or the dorsal anterior cingulate.  For example, the STN is much easier to target in Parkinson’s disease because of its compact nature and the absence of the necessity of identifying the appropriate homuncular representation within the sensori-motor region as would be the case of DBS directed at the globus pallidus interna (GPi) or ventral-intermediate nucleus of the thalamus.  Further, the compactness of axons in the vicinity of the STN may be responsible for the arguably greater clinical improvements and reduction of medications comparted to DBS in the vicinity of the GPi.  In the case of the posterior insula, it may be the closer physiological connections to the anterior cingulate where there is considerable work demonstrating that structure may be a node in the different networks involved in pain, anticipation of pain and the affective response that may be even more critical in chronic pain.  In the case of chronic pain, these details need to be worked out and the authors presented a good rationale for doing so.

Presumption 2: the relations between DBS frequency and clinical response is a simple montonically increasing function.  This presumption is the same error that arose in DBS for movement disorders, such as Parkinson’s disease.  It was presumed that low frequency excites thereby worsening symptoms and disabilities and that high frequency stimulation suppresses activity and thereby improved symptoms.   This presumption held back advances in the field for decades and the effects are still felt.  Further, the extrapolation of this presumption may account for the failure of DBS for depression.  The fact is the frequency-response relationship is far more complicated and idiosyncratic, at least reasoning from experience in Parkinson’s disease.

The authors have done an excellent job describing the pieces of wood and their arrangement to build an excellent structure.  The authors need to pay similar attention to the nails and hammers.

Author Response

We thank the reviewer for these comments. We agree that the previous version of the manuscript did not include enough discussion of the hypotheses and unknowns regarding the potential effects of electrical stimulation of the posterior insula on the activity of pain networks.  I read with interest the suggested textbook (Montgomery 2017), which covers in great depth the potential mechanisms underlying the clinical effectiveness of high-frequency DBS. 

In the previous version of the manuscript, we made the conscious choice to refrain from speculating in length on the potential neurophysiological effects of electrical stimulation of this region, because we felt that there was no sufficient literature to speculate in this regard.

We added a paragraph to discuss in more detail the hypotheses and unknowns regarding the potential effects of electrical stimulation of the posterior insula on the activity of pain networks (page 10, highlighted in yellow). We highlight that using a sensing-enabled DBS system will help to better understand the effect of different stimulation parameters on insular neurophysiology.

Regarding the two mentioned presumptions:

  • we agree that the effect of DBS cannot be solely attributed to the modulation of neurons of the implanted region. However, we do not think that this makes "DBS of the insula" a misnomer. In DBS, the electrode has to be implanted in a precise anatomical region. We refer to DBS by the region where the electrode is implanted (DBS of the STN, DBS of the GPi). Of course, the electrical stimulation interferes with the activity of neurons in the vicinity of this region, in a diameter influenced by the intensity of the stimulation. I am not aware that "DBS in the vicinity of the X" is standard terminology in the field of functional neurosurgery.
  • We agree that suggesting "low-frequency stimulation = activation" and "high-frequency stimulation = inhibition" is oversimplifying at best, and straight false at worst. We removed any sentence that evoked that model without nuance. Neverthess, when discussing the stimulation parameters that should be used in an exploratory clinical trial, we have to ground our reasoning on the available evidence, no matter how thin it is. In response to the reviewer's comment, we added more nuance in the manuscript when discussing the hypotheses and unknowns regarding the potential effects of electrical stimulation of the posterior insula on the activity of pain networks.

We thank the reviewer for his comments, which  we feel,  have helped improving the overall quality of the manuscript.

Round 2

Reviewer 2 Report

The authors’ response was disappointing. 

The authors wrote “I am not aware that "DBS in the vicinity of the X" is standard terminology in the field of functional neurosurgery.”  Nothing in science is taken as “the standard”.  If this were the case, we still would be talking about “pholgistom”.  This is an insult to very scientist whose concern is to get closer to the truth of nature.  By the authors reasoning, a bomb that goes off at the Hoover dam and causes the light to go out in LA would have to say the lights went out at the Hoover dam.  Why perpetuate terminology that only reinforces what should be easily seen as lacking face value.

The authors wrote “Nevertheless, when discussing the stimulation parameters that should be used in an exploratory clinical trial, we have to ground our reasoning on the available evidence, no matter how thin it is.”  In other words, it appears to the authors that making science a “house of cards” is acceptable.

They also wrote “Low frequency stimulation of the posterior insula was shown to elicit pain (perhaps activating pain pathways involving the posterior insula) [39], and high-frequency stimulation was shown to transiently increase temperature pain thresholds [43] (perhaps inhibiting these pain pathways). However, the effects of high-frequency electrical stimulation on neuronal firing and network dynamics remain highly debated.”  This is poor science.  It is the Fallacy of Limited Alternatives.  If a person presents only one alternative when there are others, failure to do justice to the others gives false credence to the alternative mentioned.  This does a disservice to science.  A physician would not be helping the patient by defaulting to a single diagnosis without considering the differential diagnosis.

What is particularly disappointing is that generally the paper is very strong except these flaws.  It would take so little effort to fix.  Retaining flaws when straightforward and simply remedies are available is sloppy.

Author Response

We thank the reviewer for this comment. The review rightly points out the thin scientific basis we have to predict the neurophysiological effects of electrical stimulation in the vicinity of the posterior insula. 

We recognize that an eventual trial of DBS of the insula is exploratory, with a somewhat strong basis for anatomical target, but a weak scientific basis for predicting the effect of electrical stimulation on the neuronal activity of pain networks. We modified the manuscript to stick to the facts and avoid making unsubstantiated claims. 

Regarding the terminology, we added some reference to the stimulation of the insula “and its vicinity”, as it may better reflect the neurophysiological effect of DBS.